# Analytical Solution for Forced Vibration Characteristics of Rotating Functionally Graded Blades under Rub-Impact and Base Excitation

**DOI:** 10.3390/ma15062175

**Published:** 2022-03-15

**Authors:** Tianyu Zhao, Yuxuan Wang, Xinze Cui, Xin Wang

**Affiliations:** 1Key Laboratory of Structural Dynamics of Liaoning Province, School of Science, Northeastern University, Shenyang 110819, China; zhaotianyu@mail.neu.edu.cn (T.Z.); sqwyx95@163.com (Y.W.); 2Department of Kinesiology, Shenyang Sport University, Shenyang 110102, China; cuixinze0323@163.com

**Keywords:** functionally graded material, rotating blades, rub-impact, base excitation, analytical solution

## Abstract

This paper presents an analytical investigation on the forced vibration characteristics of a rotating functionally graded material (FGM) blade subjected to rub-impact and base excitation. Based on the Kirchhoff plate theory, the rotating blade is modelled theoretically. The material properties of the FGM blade are considered to vary continuously and smoothly along the thickness direction according to a volume fraction power-law distribution. By employing Hamilton’s principle, the equations of motion are derived. Then, the Galerkin method and the small parameter perturbation method are utilized to obtain the analytical solution for the composite blade under a combined action of radial force, tangential force and displacement load. Finally, special attention is given to the effects of power-law index, rub-impact location, friction coefficient, base excitation amplitude and blade aspect ratio on the vibration characteristics of the FGM structure. The obtained results can play a role in the design of rotating FGM blades to achieve significantly improved structural performance.

## 1. Introduction

Rotating blades [1,2,3] with a low aspect ratio can compresses gas effectively; hence, they are widely used in real-world engineering applications, such as in a gas turbine or an aeroengine. In order to improve the performance of aeroengine, the radial clearance between the rotor blade and the casing needs to be as small as possible. However, the probability of rub-impact between the casing and blade tip increases with a decrease in radial clearance. A rub-impact fault may cause the complex vibration of blades and very high contact stresses that lead to blade fracture and degradation of system performance. Many major accidents have happened over an extended period. For instance, in 1973, an engine fan disintegrated in flight owing to rub-impact, as reported by the National Transportation Safety Board (NTSB). From 1994 to 1996, four F16 aircraft accidents happened due to rub-impact fault. A homemade carrier aircraft that was equipped with a WJ5A aircraft engine was grounded on account of the touch between the stator and rotor. This problem has received extensive attention from the scholars around the world.

Ma et al. [4] established improved rubbing models between a rotating blade and casing based on different methods. Liu et al. [5] focused on the dynamic responses of the whole aeroengine with a blade-casing rubbing. Hou et al. [6] studied the mechanism of a complex bifurcation behavior caused by flight maneuvers in an aircraft rub-impact rotor system with Duffing-type nonlinearity. Xiao et al. [7] developed a nonlinear dynamic model of the single-stage reciprocating compressor system with a rub-impact fault caused by subsidence, considering the piston rod’s flexibility. Tchomeni et al. [8] developed a two-dimensional model of the Navier–Stokes equations for incompressible flow for the viscous fluid motion around the spinning rotor under high fluctuations induced by unbalance, rotor–stator rub and a crack. Wang et al. [9] considered rub-impact forces, eccentricity of the rotor, and nonlinear stiffness of the armature shaft, and built a dynamic differential equation to investigate the bifurcation and chaos behavior of the locomotive traction system. Ebrahim et al. [10] investigated nonlinear dynamics due to rub-impact within tilting pad journal bearings supporting a flexible rotor. In theory, thin blades are generally established as elastic plate models. It is noted that few studies focus on the analytical analysis of forced vibrations of a rotating plate subject to the combined loads of rub-impact and base excitation.

With the rapid development of modern material science [11,12,13], a trend to substitute advanced lightweight materials in aerorotor systems has emerged. In the mid-1980s, a new concept of FGMs was first introduced by a group of Japanese scientists [14] in the context of high-performance demand of composite materials for aerospace applications. FGM is a material with continuously varying compositions from one surface to the other. Therefore, its material properties and microstructure are not uniform. In general, FGM is constructed as follows: one surface is ceramic which is designed to withstand severe external loads, such as high temperature, wear and corrosion; the other surface is attached to another material that is designed to ensure excellent toughness and thermal conductivity. The composition changes gradually along the thickness direction according to a designed law. The most significant advantages of FGM are not only improving the bonding strength, material hardness, abrasion resistance and corrosion resistance between the two materials, but also reducing the thermal stress and residual stress. Therefore, FGM is an excellent material for today’s engineering applications. It has been researched widely as FGM has excellent performances compared with traditional materials.

Shen et al. [15] presented free and forced vibration analyses for initially stressed FGM plates in a thermal environment. Kumar et al. [16] proposed two new higher-order transverse shear deformation plate theories with five variables. Singh et al. [17] investigated the buckling responses of FGM plates subjected to uniform, linear and nonlinear in-plane loads. Do et al. [18] analyzed the buckling responses of FGM plates under diverse types of thermal loadings. Wang et al. [19] analyzed the vibration of FGM beams through critical examination of midplane and neutral plane formulations. To perform further analyses, Chen et al. [20] proposed a novel FGM porous plate in which the continuous gradient in material properties based on a graded porosity offers a smooth stress distribution along the plate thickness. Yang and Zhao et al. [21,22,23,24,25,26,27,28,29] made a systematic and extensive analysis of an FGM graphene-reinforced composite structure. Li et al. [30] focused on the elastic structural stability analysis of the pressurized thin-walled FGM arches in a temperature variation field. Bourada and Bousahla et al. [31,32,33] studied the buckling and vibration of several FGM structures. Sobhy et al. [34,35] investigated the bending and wave propagation of FGM graphene-reinforced structures. To sum up, the vibration behaviors of many FGM structures have been studied. However, to the best of our knowledge, almost no study has been performed on the forced vibration of FGM plates subjected to rub-impact and base excitation.

The increasing flight speed of spacecraft has necessitated higher requirements for high-performance blades. Traditional homogeneous metal alloys no longer meet the requirement. FGM is more widely applicable in this industry due to its excellent mechanical properties, such as mitigating thermal stress, residual stress and stress concentration. Thus, the blade structure is considered as a rotating FGM plate in this paper. In addition, its forced vibration behaviors when subjected to rub-impact and base excitation need to be investigated in detail. The differential equations of motion are derived based on Hamilton’s principle. Moreover, the analytic solutions are obtained by employing the Galerkin method and the small parameter perturbation method. Finally, the effects of power-law index, rub-impact location, friction coefficient, base excitation amplitude and plate aspect ratio on the forced vibration responses are highlighted. Consequently, this paper can provide theoretical guidance and technical support for the further development and application of rotating FGM structure design.

## 2. Theoretical Formulations

### 2.1. Modeling

A rotating FGM plate model subject to rub-impact force and base excitation is established as shown in Figure 1. For the convenience of the subsequent analysis, two coordinate systems are proposed. *O*_1_-*X*_1_*Y*_1_*Z*_1_ is the fixed coordinate system, where *X*_1_ is the radial direction, *Y*_1_ is the axial direction and *Z*_1_ is the radial direction. The plate rotates at the angular velocity *Ω* along the *Y*_1_-axis. *O*-*XYZ* is the rotating coordinate system, in which the origin *O* is fixed at the corner of the plate. The angle between the *X*_1_-axis and the *X*-axis is *Ω t*. The sizes of the plate along three directions are *a*, *b* and *h*, respectively.

The presented plate is considered as a functionally gradient material structure whose material properties have the following forms:(1)$$\left\{ \begin{array}{l} E\left(z\right)=\left(E_{1}-E_{2}\right){\left(\frac{2z+h}{2h}\right)}^{n}+E_{2}\\ \upsilon \left(z\right)=\left(\upsilon_{1}-\upsilon_{2}\right){\left(\frac{2z+h}{2h}\right)}^{n}+\upsilon_{2}\\ \rho \left(z\right)=\left(\rho_{1}-\rho_{2}\right){\left(\frac{2z+h}{2h}\right)}^{n}+\rho_{2} \end{array}\right.$$
where *n* is the power-law index that dictates the material variation profile through the plate thickness. The above formulas mean that for *z* = −*h*/2, *E*(z) = *E*_2_, *υ* (z) = *υ*_2_, *ρ*(z) = *ρ*_2_, while for *z* = *h*/2, *E*(z) = *E*_1_, *υ*(z) = *υ*_1_, *ρ*(z) = *ρ*_1_. The material properties vary continuously and smoothly from *z* = −*h*/2 to *z* = *h*/2 along the thickness direction of the FGM structure.

A sine harmonic excitation along the *X*-axis direction is taken into account as the base excitation [36], expressed as:(2)$$\left\{ \begin{array}{l} u_{b}\left(t\right)=u_{0}\sin\omega_{0}t\\ {\dot{u}}_{b}\left(t\right)=u_{0}\omega_{0}\cos\omega_{0}t\\ {\ddot{u}}_{b}\left(t\right)=-u_{0}\omega_{0}^{2}\sin\omega_{0}t \end{array}\right.$$
where *u*_0_ and *ω*_0_ are the amplitude and frequency of base excitation, respectively.

The rub-impact fault consists of an impact force FD, located at the point of contact, and the resulting friction *μF_D_*, in which *μ* is the friction coefficient. The impact force, shown in Figure 2, is assumed as a segmental periodic sinusoidal pulse excitation in the form of [37]:(3)$$F_{D}(t)=\left\{ \begin{array}{ll} 0 & (n-1)T_{c}+t_{p}<t<nT_{c}\\ F_{\max}\sin\left[\frac{\pi }{t_{p}}\left(t-(n-1)T_{c}\right)\right] & (n-1)T_{c}\leq t\leq (n-1)T_{c}+t_{p} \end{array}\right.$$
where *n* = 1, 2, 3, …; *t_p_* is the impact time of one period; *T_c_* is the periodic time; *F*_max_ is the amplitude of impact force.

According to the principle of Fourier expansion, Equation (3) can be rewritten as:(4)$$F_{D}(t)=a_{0}+\sum_{k=1}^{\infty }\left(a_{k}\cos\frac{2k\pi }{T_{c}}t+b_{k}\sin\frac{2k\pi }{T_{c}}t\right)$$
where
(5)$$\left\{ \begin{array}{l} a_{0}=\frac{1}{T_{c}}\int_{0}^{t_{p}}\left[F_{\max}\sin\left(\frac{\pi }{t_{p}}t\right)\right]dt\\ a_{k}=\frac{2}{T_{c}}\int_{0}^{t_{p}}\left[F_{\max}\sin\left(\frac{\pi }{t_{p}}t\right)\cos\left(\frac{2k\pi }{T}t\right)\right]dt\\ b_{k}=\frac{2}{T_{c}}\int_{0}^{t_{p}}\left[F_{\max}\sin\left(\frac{\pi }{t_{p}}t\right)\sin\left(\frac{2k\pi }{T}t\right)\right]dt \end{array}\right.$$

### 2.2. Energy Functionals

The position vector of an arbitrary point *M* is:(6)$$r_{\mathrm{OM}}=xi+yj+wk$$
where **i**, **j** and **k** are the unit vectors of the rotating coordinate system in the *X*-axis, *Y*-axis and *Z*-axis directions, respectively; *w* is the transverse deformation of the plate.

The position relation between the rotating and fixed coordinate system is:(7)$$r_{O_{1}O}=u_{b}i$$

From this, the position vector of point *M* in the fixed coordinate system can be determined by:(8)$$r_{O_{1}M}=r_{O_{1}O}+r_{\mathrm{OM}}=\left(x+u_{b}\right)i+yj+wk$$

According to the vector and velocity relationship between the rotating and fixed coordinate system:(9)$$\left\{ \begin{array}{l} i=i_{1}\cos\Omega t-k_{1}\sin\Omega t\\ j=j_{1}\\ k=i_{1}\sin\Omega t+k_{1}\cos\Omega t \end{array}\right.\left\{ \begin{array}{l} i^{\prime }=-\Omega i_{1}\sin\Omega t-\Omega k_{1}\cos\Omega t=-\Omega k\\ j^{\prime }=0\\ k^{\prime }=\Omega i_{1}\cos\Omega t-\Omega k_{1}\sin\Omega t=\Omega i \end{array}\right.$$
in which **i**_1_, **j**_1_ and **k**_1_ are the unit vectors of fixed coordinate system, then:(10)$$\begin{array}{cc} {\dot{r}}_{O_{1}M} & =\left(x+u_{b}\right)i^{\prime }+{\dot{u}}_{b}i+wk^{\prime }+\dot{w}k\\ & =\left(w\Omega +{\dot{u}}_{b}\right)i+\left(\dot{w}-x\Omega -u_{b}\Omega \right)k \end{array}$$

The kinetic energy of the plate can be derived as:(11)$$\begin{array}{l} T_{M}=\frac{1}{2}\int_{-h/2}^{h/2}\int_{0}^{b}\int_{0}^{a}\rho {\dot{r}}_{O_{1}M}{}^{2}dxdydz\\ =\frac{1}{2}\int_{-h/2}^{h/2}\int_{0}^{b}\int_{0}^{a}\rho \left[\begin{array}{l} {\dot{u}}_{b}{}^{2}+w^{2}\Omega^{2}+2{\dot{u}}_{b}w\Omega +{\left(\frac{\partial w}{\partial t}\right)}^{2}+\\ {\left(u_{b}+x\right)}^{2}\Omega^{2}-2\frac{\partial w}{\partial t}\left(u_{b}+x\right)\Omega \end{array}\right]dxdydz \end{array}$$

Based on the Kirchhoff plate theory [38], the constitutive relations are:(12)$$\left\{ \begin{array}{l} \varepsilon_{x}=-\frac{\partial^{2}w}{\partial x^{2}}z\\ \varepsilon_{y}=-\frac{\partial^{2}w}{\partial y^{2}}z\\ \gamma_{xy}=-2\frac{\partial^{2}w}{\partial x\partial y}z \end{array}\right.,\left\{ \begin{array}{l} \sigma_{x}=-\frac{Ez}{1-\upsilon^{2}}\left(\frac{\partial^{2}w}{\partial x^{2}}+\upsilon \frac{\partial^{2}w}{\partial y^{2}}\right)\\ \sigma_{y}=-\frac{Ez}{1-\upsilon^{2}}\left(\frac{\partial^{2}w}{\partial y^{2}}+\upsilon \frac{\partial^{2}w}{\partial x^{2}}\right)\\ \tau_{xy}=-\frac{Ez}{1+\upsilon }\frac{\partial^{2}w}{\partial x\partial y} \end{array}\right.$$

The deformation potential energy of the plate is given by:

(13)$$\begin{array}{l} U_{1}=\frac{1}{2}\int_{-h/2}^{h/2}\int_{0}^{b}\int_{0}^{a}\left(\sigma_{x}\varepsilon_{x}+\sigma_{y}\varepsilon_{y}+\tau_{xy}\gamma_{xy}\right)dxdydz\\ =\frac{1}{2}\int_{-h/2}^{h/2}\int_{0}^{b}\int_{0}^{a}Dz^{2}\left[\begin{array}{l} {\left(\frac{\partial^{2}w}{\partial x^{2}}\right)}^{2}+{\left(\frac{\partial^{2}w}{\partial y^{2}}\right)}^{2}+\\ 2\upsilon \left(\frac{\partial^{2}w}{\partial x^{2}}\right)\left(\frac{\partial^{2}w}{\partial y^{2}}\right)+2\left(1-\upsilon \right){\left(\frac{\partial^{2}w}{\partial x\partial y}\right)}^{2} \end{array}\right]dxdydz \end{array}$$ where *D* = *E*/(1 − *υ*^2^).

When the instantaneous coordinates of an arbitrary point are taken as (*x*, *y*), the centrifugal force per unit volume of the plate is:(14)$$F_{1}=\rho \Omega^{2}\left(u_{b}+x\right)$$

The inertia force per unit volume of the plate is:(15)$$F_{2}=\rho {\ddot{u}}_{b}$$

According to the d’Alembert principle, the total force caused by rotation is:(16)$$F_{3}=\rho \Omega^{2}\left(u_{b}+x\right)-\rho {\ddot{u}}_{b}$$

The corresponding displacement can be calculated by:(17)$$ds-dx=\sqrt{{\left(dx\right)}^{2}+{\left(\frac{\partial w}{\partial x}dx\right)}^{2}}-dx=\frac{{\left(\frac{\partial w}{\partial x}dx\right)}^{2}}{\sqrt{{\left(dx\right)}^{2}+{\left(\frac{\partial w}{\partial x}dx\right)}^{2}}+dx}\approx \frac{1}{2}{\left(\frac{\partial w}{\partial x}\right)}^{2}dx$$

The work that is done by *F*_3_ is:(18)$${\bar{U}}_{2}=\int_{x}^{a}F_{3}\left(ds-dx\right)=\frac{1}{2}\left[\rho \Omega^{2}u_{b}\left(a-x\right)+\frac{1}{2}\rho \Omega^{2}\left(a^{2}-x^{2}\right)-\rho {\ddot{u}}_{b}\left(a-x\right)\right]{\left(\frac{\partial w}{\partial x}\right)}^{2}$$

The centrifugal potential energy of the plate is stated as:(19)$$\begin{array}{l} U_{2}=\int_{-h/2}^{h/2}\int_{0}^{b}\int_{0}^{a}{\bar{U}}_{2}dxdydz\\ =\frac{1}{2}\int_{-h/2}^{h/2}\int_{0}^{b}\int_{0}^{a}\left\{\left[\begin{array}{l} \rho \Omega^{2}u_{b}\left(a-x\right)+\frac{1}{2}\rho \Omega^{2}\left(a^{2}-x^{2}\right)\\ -\rho {\ddot{u}}_{b}\left(a-x\right) \end{array}\right]{\left(\frac{\partial w}{\partial x}\right)}^{2}\right\}dxdydz \end{array}$$

As a result, the total potential energy of the plate is:(20)$$\begin{array}{l} U_{M}=\frac{1}{2}\int_{-h/2}^{h/2}\int_{0}^{b}\int_{0}^{a}D\left\{\begin{array}{l} {\left(\frac{\partial^{2}w}{\partial x^{2}}+\frac{\partial^{2}w}{\partial y^{2}}\right)}^{2}-\\ 2\left(1-\upsilon \right)\left[\frac{\partial^{2}w}{\partial x^{2}}\frac{\partial^{2}w}{\partial y^{2}}-{\left(\frac{\partial^{2}w}{\partial x\partial y}\right)}^{2}\right] \end{array}\right\}z^{2}dxdydz\\ +\frac{1}{2}\int_{-h/2}^{h/2}\int_{0}^{b}\int_{0}^{a}\left\{\left[\begin{array}{l} \frac{1}{2}\rho \Omega^{2}\left(a^{2}-x^{2}\right)+\\ \rho \left(\Omega^{2}u_{b}-{\ddot{u}}_{b}\right)\left(a-x\right) \end{array}\right]{\left(\frac{\partial w}{\partial x}\right)}^{2}\right\}dxdydz \end{array}$$

The virtual work done by the rub-impact force is:(21)$$\delta W_{D}=q(x,y,t)\delta w$$
where
(22)$$q(x,y,t)=\mu F_{D}(t)\delta (x-x_{D})\delta (y-y_{D})$$
in which *x_D_* and *y_D_* are the rotating coordinates of an arbitrary rub-impact point *D*.

### 2.3. Governing Equations

Applying Hamilton’s principle:(23)$$\delta \int_{t_{0}}^{t_{1}}\left(T_{M}-U_{M}\right)dt+\int_{t_{0}}^{t_{1}}\int_{0}^{a}\int_{0}^{b}\delta W_{D}dxdydt=0$$
and substituting Equations (11), (20) and (21) into Equation (23) lead to the governing equation of motion, expressed as:(24)$$\begin{array}{l} \int_{-h/2}^{h/2}\left[\rho \left(w\Omega^{2}+\ddot{w}\right)\right]dz-\int_{-h/2}^{h/2}\left\{\rho \left(a-x\right)\left[\Omega^{2}\left(\frac{a+x}{2}+u_{b}\right)-{\ddot{u}}_{b}\right]\frac{\partial^{2}w}{\partial x^{2}}\right\}dz\\ -\int_{-h/2}^{h/2}\left\{D\left[\frac{\partial^{4}w}{\partial x^{4}}+\frac{\partial^{4}w}{\partial y^{4}}+\upsilon \left(\frac{\partial^{4}w}{\partial x^{2}\partial y^{2}}+\frac{\partial^{4}w}{\partial y^{2}\partial x^{2}}\right)+2\left(1-\upsilon \right)\frac{\partial^{4}w}{\partial x\partial y\partial x\partial y}\right]z^{2}\right\}dz\\ =-q(x,y,t) \end{array}$$

When the rub-impact and base excitation are ignored, Equation (30) can be given by
(25)$$\begin{array}{l} \int_{-h/2}^{h/2}\left[\rho \left(w\Omega^{2}+\ddot{w}\right)\right]dz-\frac{1}{2}\int_{-h/2}^{h/2}\left[\rho \Omega^{2}\left(a^{2}-x^{2}\right)\frac{\partial^{2}w}{\partial x^{2}}\right]dz\\ -\int_{-h/2}^{h/2}\left\{D\left[\frac{\partial^{4}w}{\partial x^{4}}+\frac{\partial^{4}w}{\partial y^{4}}+\upsilon \left(\frac{\partial^{4}w}{\partial x^{2}\partial y^{2}}+\frac{\partial^{4}w}{\partial y^{2}\partial x^{2}}\right)+2\left(1-\upsilon \right)\frac{\partial^{2}w}{\partial x\partial y}\frac{\partial^{2}w}{\partial x\partial y}\right]z^{2}\right\}dz=0 \end{array}$$

The deformation is assumed as:(26)w(x,y,t)=W(x,y)sin(ωt+ϕ)
where:
(27)$$w\left(x,y\right)=\sum_{m=1}^{M}\sum_{n=1}^{N}A_{mn}\phi_{m}\left(x\right)\phi_{n}\left(y\right)$$ in which [39]:(28)$$\phi_{m}\left(x\right)=\cosh(\alpha_{m}x)-\cos\left(\alpha_{m}x\right)-c_{m}\left[\sinh(\alpha_{m}x)-\sin\left(\alpha_{m}x\right)\right]$$
(29)$$\left\{ \begin{array}{l} \varphi_{1}\left(y\right)=1,\varphi_{2}\left(y\right)=1-\frac{2y}{b}\\ \varphi_{n}\left(y\right)=\cosh(\beta_{n}y)+\cos\left(\beta_{n}y\right)-d_{n}\left[\sinh(\beta_{n}y)+\sin\left(\beta_{n}y\right)\right] \end{array}\right.$$
where
(30)$$\left\{ \begin{array}{l} \cosh\left(\alpha_{m}a\right)\cos\left(\alpha_{m}a\right)=-1\\ c_{m}=\frac{\cos\left(\alpha_{m}a\right)+\cosh\left(\alpha_{m}a\right)}{\sin\left(\alpha_{m}a\right)+\sinh\left(\alpha_{m}a\right)}\\ m=1,2,\cdots ,M \end{array}\right.,\left\{ \begin{array}{l} \cosh\left(\beta_{n}b\right)\cos\left(\beta_{n}b\right)=1\\ d_{n}=\frac{\cos\left(\beta_{n}b\right)-\cosh\left(\beta_{n}b\right)}{\sin\left(\beta_{n}b\right)-\sinh\left(\beta_{n}b\right)}\\ n=3,4,\cdots ,N \end{array}\right.$$

Substituting Equations (32) and (33) into Equation (31), the Galerkin method leads to
(31)$$\int_{-h/2}^{h/2}\int_{0}^{b}\int_{0}^{a}\left\{\begin{array}{l} D\left(1-\upsilon \right)\left[-2\sum_{m=1}^{M}\sum_{n=1}^{N}\sum_{i=1}^{M}\sum_{j=1}^{N}A_{mn}\varphi_{m}{}^{\prime }\left(x\right)\varphi_{n}{}^{\prime }\left(y\right)\phi_{i}{}^{\prime }\left(x\right)\varphi_{j}{}^{\prime }\left(y\right)\right]z^{2}\\ -D\upsilon z^{2}\left[\begin{array}{l} \sum_{m=1}^{M}\sum_{n=1}^{N}\sum_{i=1}^{M}\sum_{j=1}^{N}A_{mn}\varphi_{m}{}^{\prime \prime }\left(x\right)\varphi_{n}\left(y\right)\phi_{i}\left(x\right)\varphi_{j}{}^{\prime \prime }\left(y\right)\\ +\sum_{m=1}^{M}\sum_{n=1}^{N}\sum_{i=1}^{M}\sum_{j=1}^{N}A_{mn}\varphi_{m}\left(x\right)\varphi_{n}{}^{\prime \prime }\left(y\right)\phi_{i}{\left(x\right)}^{\prime \prime }\varphi_{j}\left(y\right) \end{array}\right]\\ +\rho \left(\Omega^{2}-\omega^{2}\right)\sum_{m=1}^{M}\sum_{n=1}^{N}\sum_{i=1}^{M}\sum_{j=1}^{N}A_{mn}\varphi_{m}\left(x\right)\varphi_{n}\left(y\right)\phi_{i}\left(x\right)\varphi_{j}\left(y\right)\\ -\frac{1}{2}\rho \Omega^{2}\left(a^{2}-x^{2}\right)\sum_{m=1}^{M}\sum_{n=1}^{N}\sum_{i=1}^{M}\sum_{j=1}^{N}A_{mn}\varphi_{m}{}^{\prime }\left(x\right)\varphi_{n}\left(y\right)\varphi_{i}{}^{\prime }\left(x\right)\varphi_{j}\left(y\right)\\ -Dz^{2}\left[\begin{array}{l} \sum_{m=1}^{M}\sum_{n=1}^{N}\sum_{i=1}^{M}\sum_{j=1}^{N}A_{mn}\varphi_{m}{}^{\prime \prime }\left(x\right)\varphi_{n}\left(y\right)\phi_{i}{\left(x\right)}^{\prime \prime }\varphi_{j}\left(y\right)\\ +\sum_{m=1}^{M}\sum_{n=1}^{N}\sum_{i=1}^{M}\sum_{j=1}^{N}A_{mn}\varphi_{m}\left(x\right)\varphi_{n}{}^{\prime \prime }\left(y\right)\phi_{i}\left(x\right)\varphi_{j}{}^{\prime \prime }\left(y\right) \end{array}\right] \end{array}\right\}dxdydz=0$$

Eliminating the coefficient *A_mn_*, then:(32)$$\left[\begin{array}{l} E\int_{-h/2}^{h/2}\rho \Omega^{2}dz-\left(H+G-2K\right)\int_{-h/2}^{h/2}D\upsilon z^{2}dz\\ -\left(I+F+2K\right)\int_{-h/2}^{h/2}Dz^{2}dz-\frac{\Omega^{2}}{2}L\int_{-h/2}^{h/2}\rho dz \end{array}\right]-\omega^{2}E\int_{-h/2}^{h/2}\rho dz=0$$
where, **E**, **F**, **G**, **H**, **I**, **K** and **L** are in the same form of:(33)$$X=\left[\begin{array}{ccccc} X_{11} & \cdots & X_{1j} & \cdots & X_{1\left(M\times N\right)}\\ \vdots & \ddots & \vdots & & \vdots \\ X_{i1} & \cdots & X_{ij} & \cdots & X_{i\left(M\times N\right)}\\ \vdots & & \vdots & \ddots & \vdots \\ X_{\left(M\times N\right)1} & \cdots & X_{\left(M\times N\right)j} & \cdots & X_{\left(M\times N\right)\left(M\times N\right)} \end{array}\right]$$
in which *Xij* is the corresponding element in each matrix, determined by:(34)$$\left\{ \begin{array}{l} E_{\left[\left(i-1\right)N+j\right]\left[\left(m-1\right)N+n\right]}=\int_{0}^{b}\int_{0}^{a}\phi_{m}\left(x\right)\varphi_{n}\left(y\right)\phi_{i}\left(x\right)\varphi_{j}\left(y\right)dxdy\\ F_{\left[\left(i-1\right)N+j\right]\left[\left(m-1\right)N+n\right]}=\int_{0}^{b}\int_{0}^{a}\phi_{m}{}^{\prime \prime }\left(x\right)\varphi_{n}\left(y\right)\phi_{i}{}^{\prime \prime }\left(x\right)\varphi_{j}\left(y\right)dxdy\\ G_{\left[\left(i-1\right)N+j\right]\left[\left(m-1\right)N+n\right]}=\int_{0}^{b}\int_{0}^{a}\phi_{m}\left(x\right)\varphi_{n}{}^{\prime \prime }\left(y\right)\phi_{i}{}^{\prime \prime }\left(x\right)\varphi_{j}\left(y\right)dxdy\\ H_{\left[\left(i-1\right)N+j\right]\left[\left(m-1\right)N+n\right]}=\int_{0}^{b}\int_{0}^{a}\phi_{m}{}^{\prime \prime }\left(x\right)\varphi_{n}\left(y\right)\phi_{i}\left(x\right)\varphi_{j}{}^{\prime \prime }\left(y\right)dxdy\\ I_{\left[\left(i-1\right)N+j\right]\left[\left(m-1\right)N+n\right]}=\int_{0}^{b}\int_{0}^{a}\phi_{m}\left(x\right)\varphi_{n}{}^{\prime \prime }\left(y\right)\phi_{i}\left(x\right)\varphi_{j}{}^{\prime \prime }\left(y\right)dxdy\\ K_{\left[\left(i-1\right)N+j\right]\left[\left(m-1\right)N+n\right]}=\int_{0}^{b}\int_{0}^{a}\phi_{m}{}^{\prime }\left(x\right)\varphi_{n}{}^{\prime }\left(y\right)\phi_{i}{}^{\prime }\left(x\right)\varphi_{j}{}^{\prime }\left(y\right)dxdy\\ L_{\left[\left(i-1\right)N+j\right]\left[\left(m-1\right)N+n\right]}=\int_{0}^{b}\int_{0}^{a}\left(a^{2}-x^{2}\right)\phi_{m}{}^{\prime }\left(x\right)\varphi_{n}\left(y\right)\phi_{i}{}^{\prime }\left(x\right)\varphi_{j}\left(y\right)dxdy \end{array}\right.$$

The natural frequencies *ω_mn_* and coefficients *A_mn_* can be obtained by solving the standard eigenvalue problem in Equation (34).

### 2.4. Analytic Solution for Forced Vibration

The analytic solution for Equation (30) is assumed as:(35)$$w\left(x,y,t\right)=\sum_{m=1}^{M}\sum_{n=1}^{N}B_{mn}\left(t\right)W_{mn}\left(x,y\right)$$
where
(36)$$W_{mn}\left(x,y\right)=A_{mn}\phi_{m}\left(x\right)\varphi_{n}\left(y\right)$$

Applying the free vibration theory and substituting Equation (36) into Equation (30) leads to:(37)$$\frac{d^{2}B_{mn}\left(t\right)}{dt^{2}}+\omega_{mn}^{2}B_{mn}(t)+\left[{\ddot{u}}_{b}(t)-\Omega^{2}u_{b}(t)\right]\frac{C_{mn}}{M_{mn}}B_{mn}(t)=\frac{P_{mn}(t)}{M_{mn}}$$
in which
(38)$$\left\{ \begin{array}{l} P_{mn}=\int_{0}^{b}\int_{0}^{a}q(x,y,t)W_{mn}(x,y)dxdy\\ M_{mn}=\int_{-h/2}^{h/2}\int_{0}^{b}\int_{0}^{a}\rho W_{mn}^{2}(x,y)dxdydz\\ C_{mn}=\int_{-h/2}^{h/2}\int_{0}^{b}\int_{0}^{a}\rho \left(a-x\right)\frac{\partial^{2}W_{mn}(x,y)}{\partial x^{2}}W_{mn}(x,y)dxdydz \end{array}\right.$$

The base excitation and rub-impact are considered as first-order small quantities in the form of:(39)$$\left\{ \begin{array}{l} u_{b}=\varepsilon u_{0}\sin\left(\omega_{0}T_{0}\right)\\ q(x,y,t)=\varepsilon q(x,y,t) \end{array}\right.$$

The response is set as a second-order small quantity, expressed as:(40)$$B_{mn}\left(T_{0},T_{1}\right)=\varepsilon B_{mn}^{1}\left(T_{0},T_{1}\right)+\varepsilon^{2}B_{mn}^{2}\left(T_{0},T_{1}\right)$$

Substituting Equations (38) and (39) into Equation (37), the first-order equation can be derived as:(41)$$\frac{\partial^{2}B_{mn}^{1}(t)}{\partial T_{0}^{2}}+\omega_{mn}^{2}B_{mn}^{1}(t)=\frac{P_{mn}(t)}{M_{mn}}$$

Based on the forced vibration theory, the analytic solution for Equation (41) can be given by:(42)$$\begin{array}{l} B_{mn}^{1}\left(x,y,t\right)=a_{mn}\sin\left(\omega_{mn}t\right)+b_{mn}\cos\left(\omega_{mn}t\right)+\mu \frac{W_{mn}\left(x_{D},y_{D}\right)}{M_{mn}}\frac{a_{0}}{\omega_{mn}^{2}}\\ \;\;\;\;\;+\mu \frac{W_{mn}\left(x_{D},y_{D}\right)}{M_{mn}}\sum_{k=1}^{\infty }\left[\frac{a_{k}}{\omega_{mn}^{2}-\omega_{d}^{2}}\cos\left(\omega_{d}t\right)+\frac{b_{k}}{\omega_{mn}^{2}-\omega_{d}^{2}}\sin\left(\omega_{d}t\right)\right] \end{array}$$
in which *ξ* = a means that the rub-impact occurs at the edge of the plate; *a*_mn_ and *b*_mn_ are constants determined by initial conditions; and *ξ**d* = 2k*ξ*/Tc is the rub-impact frequency.

Similarly, substituting (45) and (46) into Equation (43), the second-order equation can be obtained as:(43)$$\frac{\partial^{2}B_{mn}^{2}(t)}{\partial T_{0}^{2}}+\omega_{mn}^{2}B_{mn}^{2}(t)=-2\frac{\partial^{2}B_{mn}^{1}(t)}{\partial T_{0}\partial T_{1}}+\left[\Omega^{2}u_{b}(t)-{\ddot{u}}_{b}(t)\right]\frac{C_{mn}}{M_{mn}}B_{mn}^{1}(t)$$

Then, eliminating the secular term of Equation (43), namely
(44)$$\frac{\partial^{2}B_{mn}^{1}(t)}{\partial T_{0}\partial T_{1}}=0$$
gives:(45)$$\frac{\partial^{2}B_{mn}^{2}(t)}{\partial T_{0}^{2}}+\omega_{mn}^{2}B_{mn}^{2}(t)=\left[\Omega^{2}u_{b}(t)-{\ddot{u}}_{b}(t)\right]\frac{C_{mn}}{M_{mn}}B_{mn}^{1}(t)$$

Based on the principle of linear superposition, the analytic solution for Equation (45) can be written as:(46)$$\begin{array}{l} B_{mn}^{2}(t)=AA_{mn}^{1}\cos\left[\left(\omega_{0}-\omega_{mn}\right)t\right]-AA_{mn}^{2}\cos\left[\left(\omega_{0}+\omega_{mn}\right)t\right]\\ \;\;\;\;\;\;\;\;\;\;+BB_{mn}^{1}\sin\left[\left(\omega_{0}+\omega_{mn}\right)t\right]+BB_{mn}^{2}\sin\left[\left(\omega_{0}-\omega_{mn}\right)t\right]\\ \;+CC_{mn}^{1}\sum_{k=1}^{\infty }\left\{\frac{a_{k}}{\omega_{mn}^{2}-\omega_{d}^{2}}\sin\left[\left(\omega_{0}+\omega_{d}\right)t\right]\right\}+CC_{mn}^{2}\sum_{k=1}^{\infty }\left\{\frac{a_{k}}{\omega_{mn}^{2}-\omega_{d}^{2}}\sin\left[\left(\omega_{0}-\omega_{d}\right)t\right]\right\}\\ \;+DD_{mn}^{1}\sum_{k=1}^{\infty }\left\{\frac{b_{k}}{\omega_{mn}^{2}-\omega_{d}^{2}}\cos\left[\left(\omega_{0}-\omega_{d}\right)t\right]\right\}-DD_{mn}^{2}\left\{\frac{b_{k}}{\omega_{mn}^{2}-\omega_{d}^{2}}\cos\left[\left(\omega_{0}+\omega_{mn}\right)t\right]\right\} \end{array}$$
where:(47)$$\left\{ \begin{array}{l} AA_{mn}^{1/2}=u_{0}\left(\Omega^{2}+\omega_{0}{}^{2}\right)\frac{C_{mn}}{M_{mn}}\frac{a_{mn}}{2}\frac{1}{\omega_{mn}{}^{2}-{\left(\omega_{0}-/+\omega_{mn}\right)}^{2}}\\ BB_{mn}^{1/2}=u_{0}\left(\Omega^{2}+\omega_{0}{}^{2}\right)\frac{C_{mn}}{M_{mn}}\frac{b_{mn}}{2}\frac{1}{\omega_{mn}{}^{2}-{\left(\omega_{0}+/-\omega_{mn}\right)}^{2}}\\ CC_{mn}^{1/2}=u_{0}\left(\Omega^{2}+\omega_{0}{}^{2}\right)\frac{C_{mn}}{M_{mn}}\frac{\mu S_{mn}}{2M_{mn}}\frac{1}{\omega_{mn}{}^{2}-{\left(\omega_{0}+/-\omega_{mn}\right)}^{2}}\\ DD_{mn}^{1/2}=u_{0}\left(\Omega^{2}+\omega_{0}{}^{2}\right)\frac{C_{mn}}{M_{mn}}\frac{\mu S_{mn}}{2M_{mn}}\frac{1}{\omega_{mn}{}^{2}-{\left(\omega_{0}-/+\omega_{mn}\right)}^{2}} \end{array}\right.$$

Consequently, the analytical solution for forced vibration is:(48)$$w(x,y,t)=\sum_{m=1}^{M}\sum_{n=1}^{N}\left[B_{mn}^{1}\left(t\right)W_{mn}(x,y)+B_{mn}^{2}\left(t\right)W_{mn}(x,y)\right]$$

## 3. Results and Discussion

### 3.1. Validation Study

As there are no existing solutions available in the open literature for the problem being considered, the free vibrations are investigated to validate the accuracy of the present analysis. In Table 1, the results given by Yoo [40] and Zhao [41] are provided for a direct comparison with the present results. The material and structural parameters in this example are plate length *a* = 1 m, width *b* = 1 m, thickness *h* = 0.01 m, Young’s modulus *E* = 71 GPa, mass density *ρ* = 2750 kg/m^3^, and Poisson’s ratio *υ* = 0.3.

As can be observed from Table 1, the present results agree well with those in the literature. The errors among those results are very small, which indicates that the proposed model is sufficiently accurate.

### 3.2. Forced Vibration Analysis

In this section, an analytical analysis is performed on the rotating FGM plate under rub-impact and base foundation. A detailed parametric study in graphical form is conducted to investigate the influence of volume fraction index and rotating speed on frequency field and the effects of base excitation, rub-impact and plate size on displacement fields of the FGM structure, which is made of Al and Al_2_O_3_. Unless otherwise stated, the material parameters of the plate are *E*_1_ = 70,000 MPa, *ν*_1_ = 0.317756, *ρ*_1_ = 2707 kg/m^−3^, *E*_2_ = 380,000 MPa, *ν*_2_ = 0.31, *ρ*_2_ = 3800 kg/m^−3^, *n* = 0.1; the dimension parameters are *a* = 1 m, *b* = 0.3 m, and *h* = 0.01 m; and the load parameters are *T_c_* = 2π/*Ω*, *t_p_* = 0.01 *T_c_*, *F*_max_ = 6400 N, *μ* = 0.3, *ξ* = *a*, *η* = *b*/2, *ω*_0_ = *Ω* = 300 rad/s, *u*_0_ = 0.1.

The forced vibration responses of the rotating FGM plate are presented in Figure 3. It can be seen that the vibration responses along the plate length direction have obvious irregularity, while those along plate direction are relatively uniform. In addition, it is found that the time–domain response of the FGM plate at the rubbing point (*a*, *b*/2) is periodic, while the main frequencies of its responses are in the low frequency region.

Figure 4 plots the variations of the forced vibration responses of the rotating FGM plate for different power-law indices. The results show that increasing the power-law index leads to a rise in the vibration amplitudes, while the main frequencies of the responses are less affected by the power-law index. This implies that more Al_2_O_3_ in the FGM plate can enhance the structural stiffness. Moreover, it can be found from Figure 3a and Figure 4a,b that the vibration differences along the plate length direction decrease significantly with an increase in the power-law index. This means a rise in the power-law index tends to improve the vibration stationarity of the rotating FGM plate.

Figure 5 plots the variations in the forced vibration responses of the rotating FGM plate for different base excitation amplitudes. One can find that increasing base excitation amplitudes tends to achieve higher vibration amplitudes. Besides, the base excitation can cause a high frequency vibration of the rotating FGM plate, especially for 1000 Hz and 2400 Hz. This indicates that decreasing the foundation vibration plays an important role in achieving better mechanical performance. By comparing the vibration modes from Figure 3 and Figure 5, it can be seen that the vibration differences along the plate width direction decrease with an increase in base excitation amplitudes.

In Figure 6, the variations of the forced vibration responses of the rotating FGM plate for different rub-impact locations are presented where *u*_0_ = 10. It is obvious that the vibration amplitudes of the FGM plate for different rub-impact locations differ very little. However, the vibration amplitudes corresponding to 1000 Hz increase considerably when the rub-impact location is close to the midpoint of the plate edge. This shows that the high-frequency vibrations may occur in the case of the rub-impact course near the midpoint of the plate edge. Besides, the vibration differences along the plate width direction increase markedly as the rub-impact location approaches the midpoint the of plate edge. Due to the vibration differences along the width direction, the two-dimensional plate model has more advantages than a one-dimensional beam model.

Figure 7 illustrates the variations of the forced vibration responses of the rotating FGM plate for different friction coefficients. It can be seen that the main frequencies of the responses and the vibration modes change little for the different friction coefficients. On the contrary, the vibration amplitudes increase steadily with the increase in friction coefficients, which indicates that a higher friction coefficient would exacerbate the vibrations caused by rub-impact. For the purpose of preventing damage, we can decrease the friction by reducing the surface roughness between the tip of the plate and the casing during production.

Figure 8 illustrates the variations in the forced vibration responses of the rotating FGM plate for different plate width-to-length ratios in which the plate length remains constant. One can see that the vibration amplitudes decrease obviously with the increase of the plate width-to-length ratio. Therefore, the FGM plate with a higher plate width-to-length ratio should be designed to reduce vibrations in actual engineering. Moreover, it can be told from the vibration modes that the vibration fluctuation along the plate length direction decreases with an increase in the plate width-to-length ratio. This implies that a higher plate width-to-length ratio can improve the vibration stationarity of a rotating FGM plate.

## 4. Conclusions

By employing the Kirchhoff plate theory, this paper examines a rotating FGM plate subjected to rub-impact force and base excitation. The equations of motion are derived by Hamilton’s principle. Then, the analytical solutions are obtained by adopting the Galerkin method and the small parameter perturbation method. Furthermore, the effects of the power-law index, base excitation amplitude, rub-impact location, friction coefficient and plate width-to-length ratio on vibration characteristics of the rotating FGM plate are examined in detail.

The results show: (1) decreasing the power-law index leads to a decline in the vibration amplitudes, which means that setting more Al_2_O_3_ in the FGM plate can enhance the structural stiffness; (2) the base excitation can cause larger vibration amplitudes and the generated vibrations are almost in the high frequency region; (3) the vibration amplitude corresponding to high frequency increases markedly when the rub-impact location is close to the midpoint of the plate edge; (4) a greater friction coefficient would exacerbate the vibrations caused by rub-impact; (5) and FGM plates with larger width-to-length ratios should be designed to reduce vibration and improve vibration stationarity in actual engineering.

## Figures and Tables

**Figure 1 materials-15-02175-f001:**
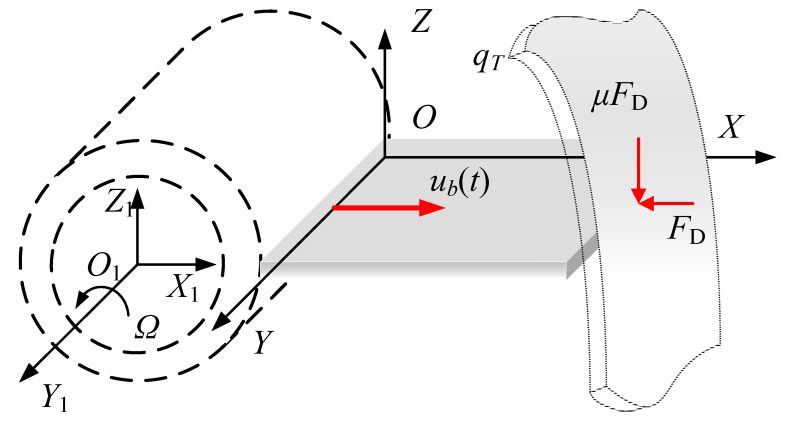
The rotating FGM plate.

**Figure 2 materials-15-02175-f002:**
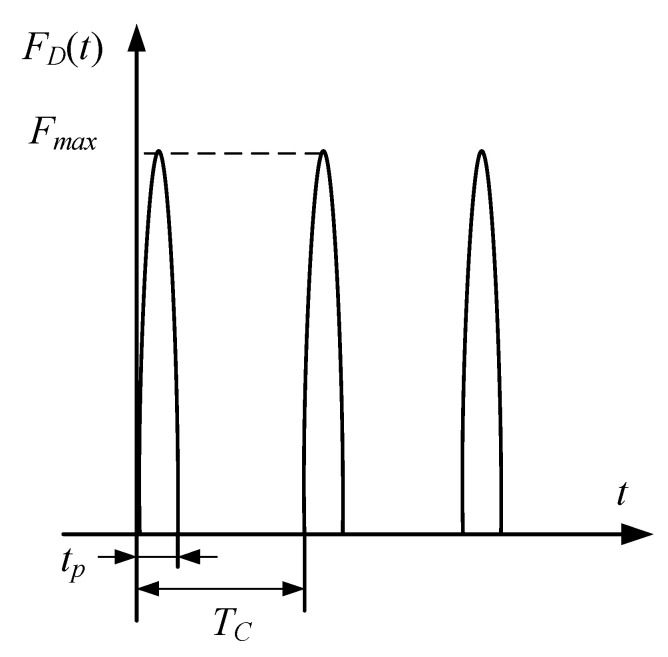
Sinusoidal pulse rub-impact force.

**Figure 3 materials-15-02175-f003:**
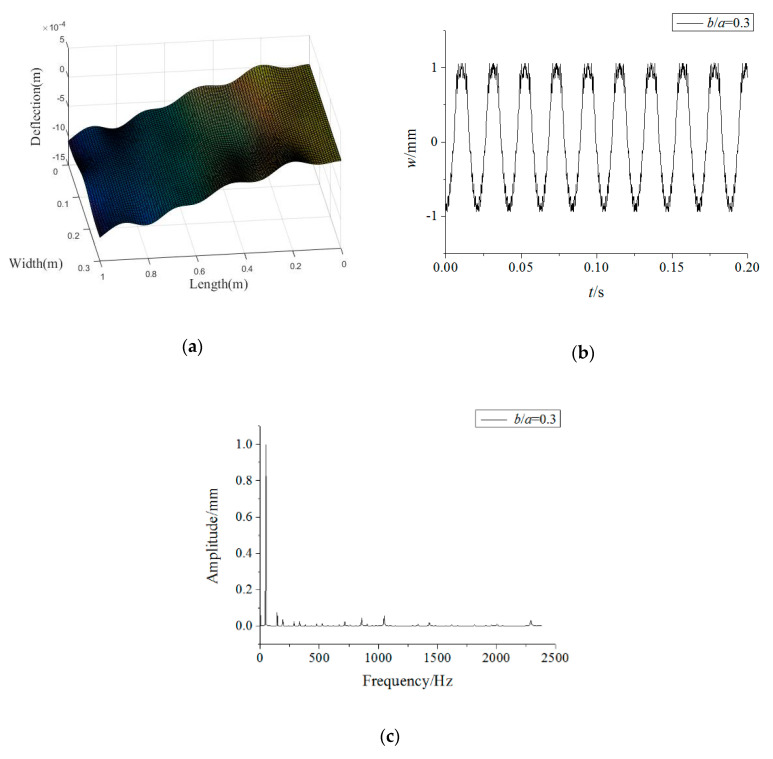
Forced vibration responses of the rotating FGM plate (**a**) vibration mode, (**b**) time-domain response, (**c**) frequency-domain response.

**Figure 4 materials-15-02175-f004:**
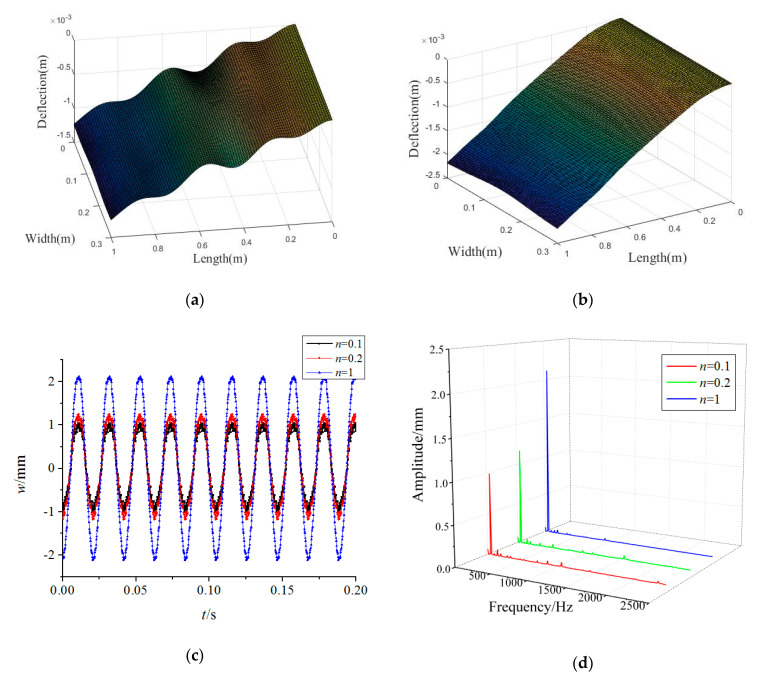
Effects of power-law indices on forced vibration responses of the rotating FGM plate (**a**) *n* = 0.2, (**b**) *n* = 1, (**c**) time-domain response, (**d**) frequency-domain response.

**Figure 5 materials-15-02175-f005:**
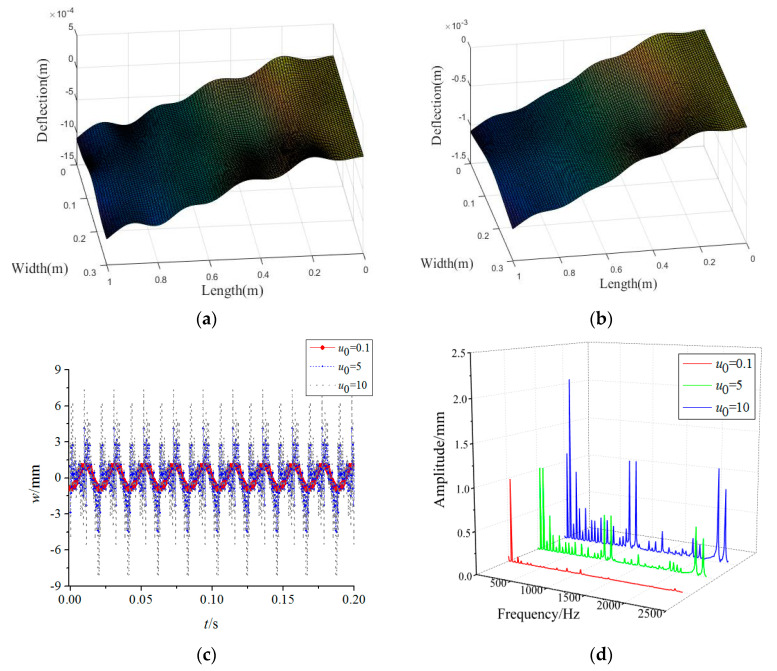
Effects of base excitation on forced vibration responses of the rotating FGM plate (**a**) *u*_0_ = 5, (**b**) *u*_0_ = 10, (**c**) time-domain response, (**d**) frequency-domain response.

**Figure 6 materials-15-02175-f006:**
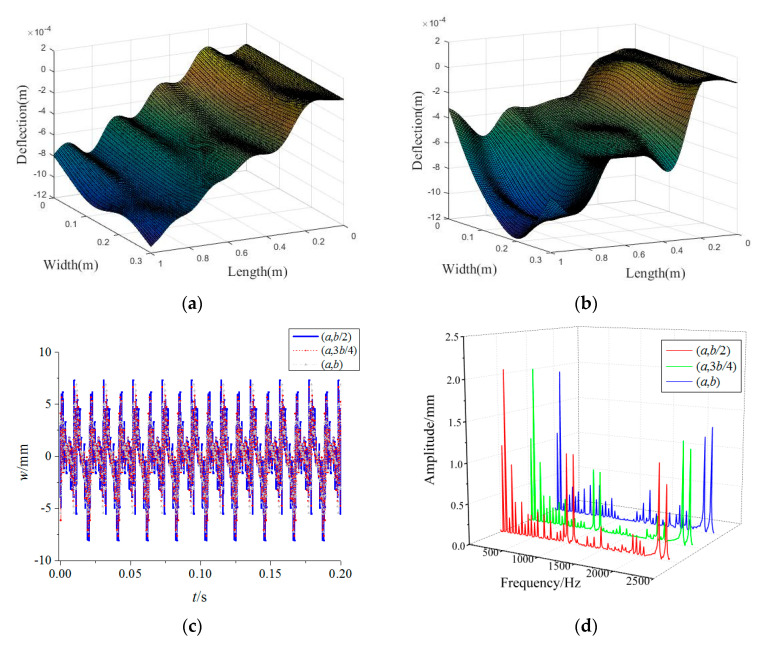
Effects of rub-impact locations on forced vibration responses of the rotating FGM plate (**a**) (*ξ*, *η*) = (*a*, 3*b*/4), (**b**) (*ξ*, *η*) = (*a*, *b*), (**c**) time-domain response, (**d**) frequency-domain response.

**Figure 7 materials-15-02175-f007:**
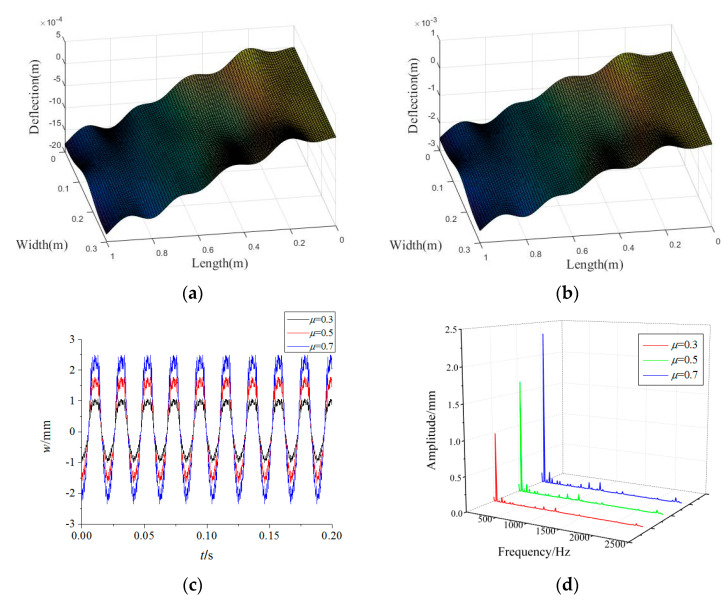
Effects of friction coefficients on forced vibration responses of the rotating FGM plate (**a**) *μ* = 0.5, (**b**) *μ* = 0.7, (**c**) time-domain response, (**d**) frequency-domain response.

**Figure 8 materials-15-02175-f008:**
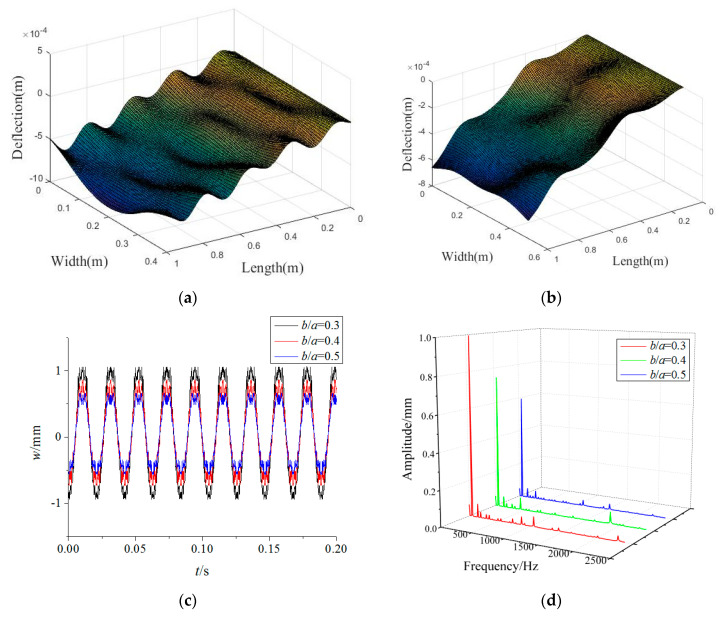
Effects of plate width-to-length ratios on forced vibration responses of the rotating FGM plate (**a**) *b*/*a* = 0.4, (**b**) *b*/*a* = 0.5, (**c**) time-domain response, (**d**) frequency-domain response.

**Table 1 materials-15-02175-t001:** Comparison of first five natural frequencies with different rotating speeds.

Dimensionless Rotating Speed	Frequency	Present	Yoo [40]	Zhao [41]
*γ* = 1	1st	3.478	3.516	3.639
	2nd	8.514	8.533	8.571
	3rd	21.325	21.520	21.469
	4th	27.208	27.353	27.194
	5th	31.013	31.206	31.068
*γ* = 2	1st	3.493	3.596	4.101
	2nd	8.517	8.551	8.755
	3rd	21.388	21.865	21.877
	4th	27.214	27.384	27.284
	5th	31.062	31.477	31.379

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
