# Peer review of "Analytical Solution for Forced Vibration Characteristics of Rotating Functionally Graded Blades under Rub-Impact and Base Excitation"

_materials, 2022, doi:10.3390/ma15062175_

Round 1
Reviewer 1 Report
Please see attached file.

Author Response
Dear Editors and Reviewers,
On behalf of my co-authors, we thank you very much for giving us an opportunity to revise our manuscript. We appreciate editors and reviewers very much for their positive and constructive comments and suggestions on our manuscript entitled “Analytical solution for forced vibration characteristics of a rotating functionally graded plate under rub-impact and base excitation”.
We have studied reviewer’s comments carefully and have tried our best to revise our manuscript according to the comments. The revision is marked in red in the manuscript. The detail point-to-point reply is listed at the bottom of this letter.
Once again, we would like to express our great appreciation to you and the reviewer for comments on our paper. Looking forward to hearing from you.
Thank you and best regards.
Yours sincerely,
Tian Yu Zhao
Northeastern University
Reviewer: 1
This paper studies application of Galerkin’s approach and Hamilton’s principle to forced vibration analysis of a functionally graded blade, in which the material properties are assumed variable along the thickness direction. Perturbation technique is used to derive solution. The effect of significant parameters such as in-homogeneous index, friction coefficient and some geometric parameters is studied on the responses. The paper needs a revision before reconsideration. The reviewer comments are suggested as follows:
- Application of the proposed model and used analysis is not clear for reviewer. Authors should present some application of the proposed model.
Reply: “Rotating blades with low aspect ratio can compresses the gas effectively and they are widely applied in the actual engineering, such as in a gas turbine, an aeroengine, et. al.” has been added in the introduction.
- More comments on the novelties and main contributions of the present paper is needed.
Reply: “In theory, thin blades are generally established as elastic plate models. It is noted that few studies focus on the analytical analysis of forced vibrations of a rotating plate subject to combined loads of rub-impact and base excitation.” and “The increasing flight speed of spacecraft has put forward higher requirements on the high-performance blades. Traditional homogeneous metal alloy no longer meets the requirement. FGM could been more widely used in this industry due to its excellent mechanical properties, such as mitigating thermal stress, residual stress and stress concentration.” have been added in the introduction.
- There are some new works on the new materials and new production applicable in engineering equipment. Some works for improvement of Introduction is suggested as follows: The introduction section should be updated with studying and citation of new papers such as: Aerospace science and technology, 111, 106534. doi: 10.1016/j.ast.2021.106534; Tribology international, 164. doi: 10.1016/j.triboint.2021.107206. There are some works relate to Manouchehr Mohammad Hosseini Mirzaei on the functionally graded blade that should be studied in Introduction.
Reply: The mentioned references have been cited in Reference [1-3].
There are some errors in the paper. A review on the paper is required.
Reply: The paper has been double checked. The revised has been marked in red.
Some relations need references for tracing.
Reply: The references have been added before Eq. (3, 12, 34).
A comparative study is required for verification of the results.
Reply: The comparative study is added in section 3.1.
The discussion is not informative. It should be enriched with addition of some more important conclusions
Reply: The Section 3 and Section 4 have been revised as marked in red.
The results and discussion are very briefly organized. An extended result and discussion are required.
Reply: The results and discussion have been extended as marked in red.
Reviewer: 2
This paper studied forced vibration analysis of a rotating blade made from functionally graded materials subjected to rub-impact and base excitation. The kinematic relations are developed based on Kirchhoff plate theory through Hamilton’s principle. The paper needs a revision before reconsideration. The reviewer comments are suggested as follows:
Although the title, materials and methods of the present paper are very interesting, however the main novelties of it are not clear for reviewers. Authors are encouraged to add more comments on the novelties and main contributions of the present paper in Abstract and last paragraph of Introduction section.
Reply: “In theory, thin blades are generally established as elastic plate models. It is noted that few studies focus on the analytical analysis of forced vibrations of a rotating plate subject to combined loads of rub-impact and base excitation.” and “The increasing flight speed of spacecraft has put forward higher requirements on the high-performance blades. Traditional homogeneous metal alloy no longer meets the requirement. FGM could been more widely used in this industry due to its excellent mechanical properties, such as mitigating thermal stress, residual stress and stress concentration.” have been added in the introduction.
What is application of the proposed models? Authors are suggested to provide some technical expressions on the application of the proposed model and needing to this new finding.
Reply: “Rotating blades with low aspect ratio can compresses the gas effectively and they are widely applied in the actual engineering, such as in a gas turbine, an aeroengine, et. al.” has been added in the introduction.
Some relations need references for tracing.
Reply: The references have been added before Eq. (3, 12, 34).
All variables should be defined after first appearance in the text.
Reply: All variables have been defined at their first appearance.
Numerical results and discussion should be corrected with addition of physical meaning of the outputs.
Reply: The Section 3 and Section 4 have been revised as marked in red.
The introduction section should be updated with studying and citation of new papers such as: Materials characterization, 171. doi: 10.1016/j.matchar.2020.110732; Engineering structures, 243, 112645. doi: 10.1016/j.engstruct.2021.112645; Aerospace science and technology, 117. doi: 10.1016/j.ast.2021.106937
Reply: The mentioned references have been cited in Reference [11-13].
Reviewer: 3
The problem of the research for homogeneous blades has long been solved. But the wide usage of functional coatings makes it necessary to study their influence on the dynamics of the blades and, in particular, their vibration response under rub-impact excitation.
The scientific novelty of the presented research and its significance is not clear because the overview of research on the topic is presented descriptively without an analysis of existing methods. Therefore, the authors should improve the introduction section. The method is of scientific interest, but the style of presentation also can be improved.
Reply: “In theory, thin blades are generally established as elastic plate models. It is noted that few studies focus on the analytical analysis of forced vibrations of a rotating plate subject to combined loads of rub-impact and base excitation.” and “The increasing flight speed of spacecraft has put forward higher requirements on the high-performance blades. Traditional homogeneous metal alloy no longer meets the requirement. FGM could been more widely used in this industry due to its excellent mechanical properties, such as mitigating thermal stress, residual stress and stress concentration.” have been added in the introduction.
Part of the formulas should be omitted or presented in a more compact form.
Reply: Several formulas have been presented in a compact form, such as Eq. (53).
The purpose of the paper is the improvement of the analytical solution for the forced vibration characteristics of the FG blades under the action of the centrifugal load and rub-impact. At the same time, the article contains detailed results of the influence of the FGM structure parameter, rub-impact location, friction coefficient, etc. on the vibration response of the plate. These results may form the basis of a new article by the authors, in our point of view. Hence, Section 3.2 must be revisited or sufficiently shorted.
Reply: The Section 3 and Section 4 have been revised as marked in red.
English in the paper needs to be substantially improved. There are a lot of editorial errors throughout the manuscript.
Reply: The paper has been doubled checked. The editorial errors have been revised.
Reviewer: 4
The paper presents an analytical solution for forced vibration characteristics of rotating functionally graded blades under rub-impact and base excitation. According to the reviewer’s opinion, the paper is well-structured and clear. The topic is interesting and falls within the aim of the journal. In addition, the results are well-presented and could be helpful to further develop the same topic. Therefore, the paper can be accepted for publication in the current form.
Reply: Thank you for your comment.
Reviewer: 5
No comments
Reviewer 2 Report
Reviewer comments on the paper “Analytical solution for forced vibration characteristics of rotating functionally graded blades under rub-impact and base excitation”
This paper studied forced vibration analysis of a rotating blade made from functionally graded materials subjected to rub-impact and base excitation. The kinematic relations are developed based on Kirchhoff plate theory through Hamilton’s principle. The paper needs a revision before reconsideration. The reviewer comments are suggested as follows:
Although the title, materials and methods of the present paper are very interesting, however the main novelties of it are not clear for reviewers. Authors are encouraged to add more comments on the novelties and main contributions of the present paper in Abstract and last paragraph of Introduction section.
What is application of the proposed models? Authors are suggested to provide some technical expressions on the application of the proposed model and needing to this new finding.
Some relations need references for tracing.
All variables should be defined after first appearance in the text.
Numerical results and discussion should be corrected with addition of physical meaning of the outputs.
The introduction section should be updated with studying and citation of new papers such as: Materials characterization, 171. doi: 10.1016/j.matchar.2020.110732; Engineering structures, 243, 112645. doi: 10.1016/j.engstruct.2021.112645; Aerospace science and technology, 117. doi: 10.1016/j.ast.2021.106937
Author Response

(The authors gave the same response as above.)

Reviewer 3 Report
The problem of the research for homogeneous blades has long been solved. But, the wide usage of functional coatings makes it necessary to study their influence on the dynamics of the blades and, in particular, their vibration response under rub-impact excitation.
The scientific novelty of the presented research and its significance is not clear because the overview of research on the topic is presented descriptively without an analysis of existing methods. Therefore, the authors should improve the introduction section. The method is of scientific interest, but the style of presentation also can be improved. Part of the formulas should be omitted or presented in a more compact form. The purpose of the paper is the improvement of the analytical solution for the forced vibration characteristics of the FG blades under the action of the centrifugal load and rub-impact. At the same time, the article contains detailed results of the influence of the FGM structure parameter, rub-impact location, friction coefficient, etc. on the vibration response of the plate. These results may form the basis of a new article by the authors, in our point of view. Hence, Section 3.2 must be revisited or sufficiently shorted. English in the paper needs to be substantially improved. There are a lot of editorial errors throughout the manuscript. In page 2, line 82 is wrong reference [25-43], instead [35-43]. The authors are strongly encouraged to have this article carefully proofread. Also, I would like to suggest the authors re-plotting figures with the same style of the X- and Y-labels and in Table 1 highlight the errors.The paper contains a high level of self-citation and one-author reference.
Author Response

(The authors gave the same response as above.)

Reviewer 4 Report
The paper presents an analytical solution for forced vibration characteristics of rotating functionally graded blades under rub-impact and base excitation. According to the reviewer’s opinion, the paper is well-structured and clear. The topic is interesting and falls within the aim of the journal. In addition, the results are well-presented and could be helpful to further develop the same topic. Therefore, the paper can be accepted for publication in the current form.
Author Response

(The authors gave the same response as above.)

Reviewer 5 Report
No comments
Author Response

(The authors gave the same response as above.)

Round 2
Reviewer 1 Report
The revised paper is suggested for publication.
Author Response
Many thanks